



# Radar Equivalent Snowpack: reducing the number of snow layers while retaining its microwave properties and bulk snow mass

Julien Meloche[1], Nicolas R. Leroux[2], Benoit Montpetit[1], Vincent Vionnet[2], and Chris Derksen[1]

[1]Climate Research Division, Environment and Climate Change Canada, Ontario, Canada
[2]Meteorological Research Division, Environment and Climate Change Canada, Quebec, Canada

**Correspondence:** Julien Meloche (julien.meloche@ec.gc.ca)

**Abstract.** Snow water equivalent (SWE) retrieval from Ku-band radar measurements is possible with complex retrieval algorithms involving prior information on the snowpack microstructure and a microwave radiative transfer model to link backscatter measurements to snow properties. A key variable in a retrieval is the number of snow layers, with more complex layering yielding richer information but at increased computational cost. Here, we show the capabilities of a new method to simplify a complex multilayered snowpack to less than or equal to 3 layers, while preserving the microwave scattering behavior of the snowpack and conserving the bulk snow water equivalent. The method is based on a K-means clustering algorithm to group the snow layers based on the extinction coefficient and the height of the layer. Then, a weighted average using the extinction coefficient and the thickness was applied to the snow properties. We evaluated our method using snow properties from simulations of the SVS-2/Crocus physical snow model at 11 sites spanning a large variety of climates across the world and the Snow Microwave Radiative Transfer model to calculate backscatter at 17.25 GHz. Grouping and averaging snow stratigraphy into 3 layers effectively reproduced the total snowpack backscatter of multi-layered snowpacks with overall root mean squared error = 0.5 dB and $R^2$ = 0.98. Using this methodology, SWE retrievals can be applied to simplified snowpacks, while maintaining similar scattering behavior, without compromising the modeled snowpack properties. Reduction in the mathematical complexity of SWE retrieval cost functions and reduction in computation of up to 80% can be gained by using fewer layers in the SWE retrieval.

## 1 Introduction

Snow Water Equivalent (SWE) is a key element of the hydrological cycle and an important component of the surface energy balance, so it must be well-represented in environmental prediction systems. Because conventional SWE observations are exceptionally sparse, new spaceborne radar missions to deliver SWE information are under development, such as the Canadian Terrestrial Snow Mass Mission (TSMM, Derksen et al., 2021; Tsang et al., 2022). A state-of-the-art SWE retrieval from the Ku-band radar measurements delivered by a mission like TSMM requires a radiative transfer model (RTM) to link snow properties to backscatter (Saberi et al., 2021; Zhu et al., 2021; Pan et al., 2017, 2023). Snow properties including layer thickness, density, temperature, and microstructure (e.g., specific surface area) are necessary to properly model the microwave signal with a RTM. Prior, or *first-guess* information on layered snow properties is needed to constrain retrievals (Merkouriadi et al., 2021; Durand



et al., 2024). This information can come from vertical snowpack measurements, either manually, or using an instrument like
a high-resolution snow penetrometer (SMP, Proksch et al., 2015). More typically, observations are unavailable, so snowpack
information must come from physical snow models that provide multi-layered snow properties based on meteorological forcing
data.

Detailed physical snow models like Crocus (Vionnet et al., 2012) and SNOWPACK (Bartelt and Lehning, 2002; Lehning

et al., 2002) are one-dimensional multi-layer physical schemes that can model the evolution of the snowpack (including its mi-
crostructure) by taking into account energy exchange between the snow, the atmosphere, and the soil based on meteorological
inputs. In Crocus, the snowpack is vertically discretized on a finite element grid with specific rules to allow the snowpack layer-
ing to evolve dynamically from new precipitation, compaction and/or metamorphism. One such rule is the dynamic attribution
of the number of layers and thicknesses to simulate the snowpack layering. The minimum number is 3 layers but the maximum

is 50 (Vionnet et al., 2012). Setting a large number of layers can improve dynamic physical processes within the snowpack,
such as heat and mass fluxes within the snowpack, and spring snowmelt initiation (Cristea et al., 2022) or the identification of
weak layers in the context of avalanche hazard forecasting (Morin et al., 2020), but can prove challenging for remote sensing
applications.

Some algorithms couple a physical snow model and a snow RTM to retrieve SWE using microwave remote sensing data

(Langlois et al., 2012; Larue et al., 2018; Singh et al., 2024). Snow RTMs can model the radar backscatter using snow param-
eters from complex layered snowpacks. In a SWE retrieval like Pan et al. (2017), the SWE (depth and density) of the different
layers is estimated by minimizing the difference between the modeled and measured backscatter. However, larger numbers of
layers increase the complexity of the retrieval by increasing the number of variables in the cost function. Also, most RTMs
solve the radiative transfer equation based on the discrete ordinate and eigenvalue method (Picard et al., 2004), which dis-

cretizes the radiative transfer equation and solves a nonhomogenous system of linear equations based on the number of layers.
Increasing the number of snow layers thus increases the computational cost at many levels within the retrieval algorithm. This
is why current retrievals typically employ a one or two-layer model (Saberi et al., 2021; Durand et al., 2024; Pan et al., 2017).
However, neglecting stratigraphy by using a small number of layers model reduces the performance of the retrieval (Durand
et al., 2011) because layering strongly influences the backscattering properties of snow (Rutter et al., 2016). A one-layer model

oversimplifies the scattering behavior of the snowpack and so is not adequate in most cases (Rutter et al., 2019; Meloche et al.,
2024; Montpetit et al., 2024). For this reason, a two or three-layer model provides notably better SWE retrievals (Pan et al.,
2017).

To reduce the number of layers, a mass or thickness weighted average is commonly used to average all properties of the
snowpack (Durand et al., 2011) and conserve snow mass (i.e., SWE). Singh et al. (2024) applied the same logic in averaging

a multi-layered snowpack into a two-layered snowpack, and chose the height that corresponded to the maximum change in
density to split the snowpack into two layers. Other SWE retrievals (Saberi et al., 2021; Pan et al., 2017) focused on arctic
snowpack by setting the initial two-layer snowpack from well-documented layer properties (e.g., wind slab and depth hoar for
Arctic snowpacks) (Rutter et al., 2019; Vargel et al., 2020; Derksen et al., 2009, 2012). For assimilation of passive microwave
data, Larue et al. (2018) used a detailed physical snow model (Crocus) coupled with a RTM with a limit of 15 layers as a





compromise between accuracy and computation time. If a method that reduces the number of layers while preserving SWE and microwave scattering exists, the accuracy would not be compromised and the computation time reduced. To our knowledge, a robust method does not exist to effectively reduce the number of layers of a given snowpack while minimizing changes in scattering properties.

The goal of this paper is to develop a simple algorithm to convert a multi-layered snowpack with a large number of layers

(20-50) into a simplified snowpack (2-3 layers) that preserves its snow mass and scattering behavior, thereby improving the computational cost with minimal impact on performance. The method should preserve backscatter within 1 dB, since it is the calibration uncertainty of most synthetic aperture radar (Schmidt et al., 2018). To evaluate our method, we test it on multi-layered Crocus simulations at 11 sites across various snow climates over multiple seasons. From the 50-layer simulations (maximum layering), we compare various methods to obtain a "radar equivalent" snowpack by evaluating differences in snow

mass and simulated backscatter using the Snow Microwave Radiative Transfer Model (SMRT, Picard et al., 2018).

## 2 Methods

### 2.1 Study Site and Data

A total of 11 sites were selected to cover a wide range of snowpack conditions and climates, and meteorological forcing and evaluation datasets were previously published. Amongst the 11 sites, 6 are in mountain environments (Col de Porte, Kühtai and

Weissfluhjoch in the European Alps, Reynolds Mountain, Senator Beck and Swamp Angel are in the Western USA); 2 are in tundra environments (Bylot and Trail Valley Creek in the Canadian high Arctic), 2 are in taiga environments (Umiujaq in the Canadian Boreal Forest and Sodankylä in the Finnish Boreal forest) and 1 is in a Maritime climate (Sapporo in Japan). Details of each dataset are shown in Table 1.

**Table 1.** Overview of sites used to evaluate Crocus snowpack layering reduction methods.

| Site | Code | Source | Time Period | Lat. (°) | Lon. (°) | Elev. (m) | Country | Snow cover |
|------|------|--------|-------------|----------|----------|-----------|---------|------------|
| Bylot | BYL | Domine et al. (2021) | 2014-2019 | 73.15 | -80.00 | 22 | Canada | Arctic |
| Col de Porte | CDP | Menard and Essery (2019) | 1994-2014 | 45.30 | 5.77 | 1325 | France | Alpine |
| Kühtai | KUT | Krajči et al. (2017) | 1990-2013 | 47.21 | 11.01 | 1920 | Austria | Alpine |
| Reynolds Mountain | RME | Menard and Essery (2019) | 1988-2008 | 43.19 | -116.78 | 2060 | USA | Alpine |
| Sapporo | SAP | Menard and Essery (2019) | 2005-2015 | 43.08 | 141.34 | 15 | Japan | Maritime |
| Senator Beck | SNB | Menard and Essery (2019) | 2005-2015 | 37.91 | -107.73 | 3714 | USA | Alpine |
| Sodankylä | SOD | Menard and Essery (2019) | 2007-2014 | 67.37 | 26.93 | 179 | Finland | Taïga |
| Swamp Angel | SWA | Menard and Essery (2019) | 2005-2015 | 37.91 | -107.71 | 3371 | USA | Alpine |
| Trail Valley Creek | TVC | Tutton et al. (2024) | 2013-2018 | 68.75 | -133.5 | 91 | Canada | Arctic |
| Umiujaq | UMQ | Lackner et al. (2023) | 2012-2020 | 56.56 | -76.48 | 130 | Canada | Taïga |
| Weissfluhjoch | WFJ | Menard and Essery (2019) | 1996-2016 | 46.83 | 9.81 | 2536 | Switzerland | Alpine |



## 2.2 SVS2-CROCUS

The snowpack model Crocus (Brun et al., 1992; Lafaysse et al., 2017; Vionnet et al., 2012) was used to simulate the evolution of the snowpack properties, i.e. number of layers, and thickness, density, liquid water content, temperature, and specific surface area for each layer specifically. The version of Crocus used in this study is implemented into the Soil, Vegetation and Snow version 2 (SVS-2) land surface scheme (Garnaud et al., 2019; Vionnet et al., 2022; Woolley et al., 2024). The snowpack model is coupled to a multi-layered soil model that includes soil freezing and thawing (Boone et al., 2000). SVS-2 is an improvement

to the SVS land surface scheme (Alavi et al., 2016; Husain et al., 2016; Leonardini et al., 2021) used at Environment and Climate Change Canada for hydrological prediction. For the simulations at the Arctic sites (Bylot and Trail Valley Creek), the Arctic version of Crocus (Royer et al., 2021b; Woolley et al., 2024) was used. This version improves the simulation of the wind slab properties and includes the impact of basal vegetation on the snowpack properties.

At each site, simulations were run with a maximum number of 50 snow layers. Table A1 summarizes the options of Crocus

for each physical process and the snow aging parameters (Gaillard et al., 2024) used for the simulations at the different sites. At the two arctic sites, the polar vegetation height was set to 10 cm (Woolley et al., 2024) and for taiga sites, it was set to 50 cm and 20 cm for the UMQ and SOD respectively. The model uses a time step of 10 minutes and the meteorological forcing is provided hourly.

## 2.3 Snow Microwave Radiative Transfer (SMRT)

SMRT is a multi-layer snow radiative transfer model that can compute backscatter in the microwave range. It considers each snow layer as a homogeneous random medium composed of air and ice and solves the radiative transfer equation for a multi-layered snowpack (Picard et al., 2018). Each snow layer is represented by temperature, density, thickness and microstructure parameters, all of which are provided by Crocus. One key component of SMRT is determining the scattering and absorption coefficients ($\kappa_\mathrm{s}$, $\kappa_\mathrm{a}$) of each layer. These coefficients dictate the radar scattering and absorbing behavior of the medium (snow).

For snow, this generally implies that scattering ($\kappa_\mathrm{s}$) dominates for dry snow, and absorption ($\kappa_\mathrm{a}$) dominates for wet snow. The extinction coefficient ($\kappa_\mathrm{e} = \kappa_\mathrm{s} + \kappa_\mathrm{a}$) characterizes the interaction within the medium by accounting for both coefficients and is a key parameter of the snow layer reduction algorithm presented in this paper. Multiple formulations of the coefficients can be used depending on the electromagnetic model, but here we focus on the Improved Born approximation (IBA, Mätzler, 1998) implemented in SMRT. The choice of the electromagnetic model does not influence the final result of this method, only the

way the $\kappa_\mathrm{e}$ is calculated.

The phase function of snow in the 1-2 frame (e.g. ice particles (medium 2) in an air matrix (medium 1)) is defined by:

$$p_{\text{1-2 frame}}(\vartheta,\varphi) = \phi_\mathrm{i}(1-\phi_\mathrm{i})(\epsilon_2-\epsilon_1)^2\,Y^2(\epsilon_2,\epsilon_1)k_0^4\,M(|k_\mathrm{d}|)\sin^2\chi \tag{1}$$

where $k_0$ is the wavenumber in free space, $\phi_\mathrm{i}$ is the volume fraction of the scattering constituent (ice) described $\phi_\mathrm{i} = \rho_\mathrm{snow}/\rho_\mathrm{ice}$ with $\rho_\mathrm{snow}$ the snow density (kg m$^{-3}$) and $\rho_\mathrm{ice}$ the pure ice density (kg m$^{-3}$), $\epsilon$ is the relative permittivity of both media and $\chi$

is the polarisation angle define from $\sin^2\chi = 1 - \sin^2\vartheta\cos^2\varphi$ for the scattering ($\vartheta$) and the incident ($\varphi$) direction. The mean





square field ratio ($Y^2$) accounts for the difference in the electric field between the background and scattering medium (see Picard et al. (2018) for equations). The microstructure term ($M(|k_d|)$) is defined by the Fourier transform of the correlation function of the medium (ACF). Here, we used the exponential model (Mätzler, 2002)) where the ACF is characterized by a correlation length ($l_{mw}$) estimated by the specific surface area (SSA) and the snow density. More details on the ACF and $M(|k_d|)$ can be found in Picard et al. (2018).

From the phase function, we define $p_{11} = p_{\text{1-2 frame}}(\vartheta, \varphi = \pi/2)$ and $p_{22} = p_{\text{1-2 frame}}(\vartheta, \varphi = \pi)$ . The $\kappa_s$ can be calculated from the following equation:

$$\kappa_s = \pi \int_0^\pi [\, p_{11}(\vartheta) + p_{22}(\vartheta)\, ] \, d\vartheta \tag{2}$$

where $p_{11} = p_{\text{1-2 frame}}(\vartheta, \varphi = \pi/2)$ and $p_{22} = p_{\text{1-2 frame}}(\vartheta, \varphi = \pi)$ are defined from the phase function. The $\kappa_a$ is defined by:

$$\kappa_a = 2k_0 \Im(\sqrt{\epsilon_{\text{eff}}}) \tag{3}$$

where $\epsilon_{\text{eff}}$ is the effective permittivity using the Polden-van Santen general mixing formula (Shivola, 1999).

In addition to the simulations using the full number of Crocus snow layers, SMRT was also used to simulate the backscatter of the snowpack to evaluate our averaging methods (Section 2.4). Crocus provides the layered snow density and SSA to estimate the microwave grain size with :

$$l_{mw} = Kl_p = K \frac{4(1 - \phi_i)}{\rho_{\text{ice}} \, SSA} \tag{4}$$

where $K$ is the polydispersity of the microstructure. The polydispersity was assumed to be 1 for all grain types in this experiment but is presented in the equation because it strongly influences scattering (Picard et al., 2022). Future work could include a $K$ for different Crocus grain types.

Simulations were performed first at the high Ku-band (17.25GHz), the TSMM frequency that is the most sensitive to volume scattering (Derksen et al., 2021) but later the frequency range, X to Ka-band, was also investigated. VV polarisation was the focus since HH will not be measured as part of TSMM (Derksen et al., 2021). A simple absorber for the background was used, i.e. no scattering from the ground is assumed, to only obtain the snow contribution to the modeled backscatter. The incident angle was set to 35 ° as a typical median value for SAR sensors such as TSMM. The snow layer interfaces were assumed to be flat.

## 2.4 Algorithm

The algorithm aims at reducing the number of snow layers to 2 or 3 relevant layers while preserving SWE and scattering behavior with respect to the reference snowpack (defined as the 50-layer Crocus simulations). The main goal is to develop a robust method to aggregate and average snow layer properties in a microwave SWE retrieval context. Figure 1 shows the general methodology and the different explored options to obtain a radar equivalent snowpack with reduced layers.

The grouping of layers was done in two ways. The first method (referred to equal thickness) aggregated layers based on the normalized height ($h_{\text{norm}}$). For an averaged two-layer snowpack, the top half of the snowpack, i.e all the layers with $h_{\text{norm}} >$



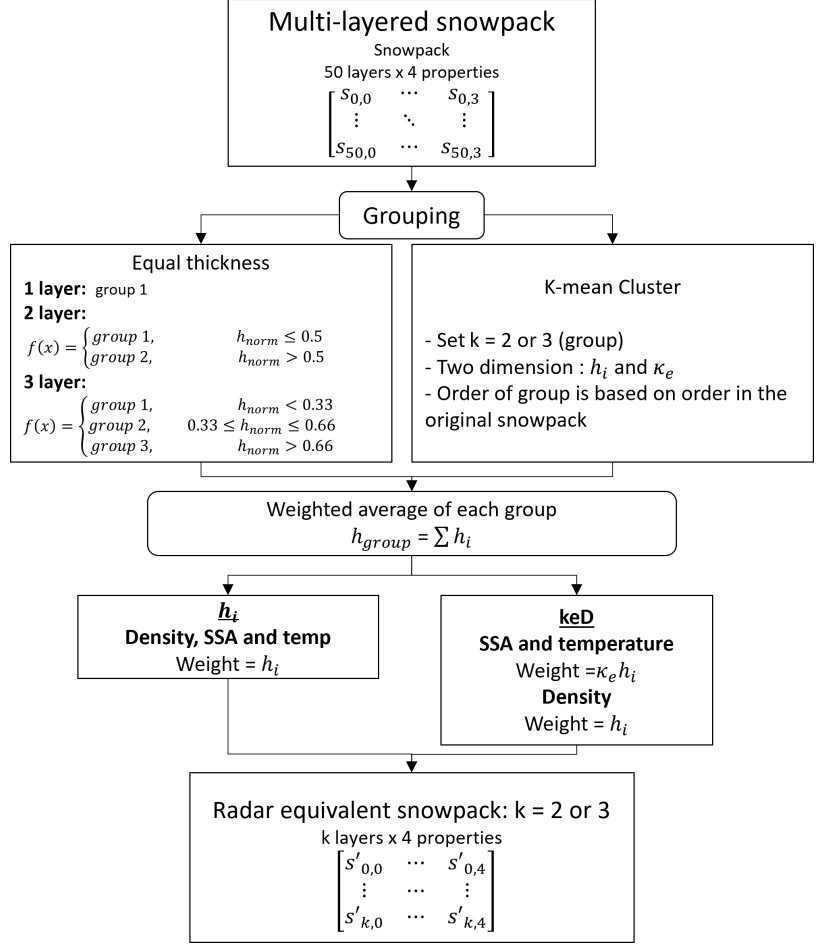

**Figure 1.** Schematic of the methodology. Necessary steps in the snowpack simplification are shown vertically and the different options (grouping and averaging) are shown horizontally where $h_{\mathrm{norm}}$ is the normalized height and $h_i$ is the layer thickness. The 4 snow properties considered are: thickness, density, temperature and SSA.

0.5 would be aggregated into a first layer and the bottom half (layer with $h_{\mathrm{norm}} \leq 0.5$) into the second layer. This method is a basic way to obtain an equivalent snowpack with layers of equal thickness. The 1-layer was created by grouping all layers into 1 group. The second method was based on K-means clustering (Ikotun et al., 2023), which finds groups based on the parameter

space. In our case, the parameter space is the extinction coefficient ($\kappa_e$) of each layer and the respective layer height in the snowpack. Layers with strong extinction coefficients (high $\kappa_e$) will have strong interaction with the incident wave. A layer with a low extinction coefficient will be practically transparent to the incident wave. This method creates groups of layers based on their microwave properties ($\kappa_e$) and their location in the snowpack.

       Once the grouping of layers is done with the K-means method, the thickness of each group ($h_{\mathrm{group}}$) is calculated by the

sum of all the individual layer thicknesses ($h_i$) within that group. For both grouping methods, the snowpack layer properties





(density, temperature and SSA) were averaged. We investigated three different ways of averaging: 1) a weighted average based on $h_i$ (referred to as $h_i$) and 2) a weighted average based on the optical thickness ($\tau = \kappa_e\, h_i$) (Zhu et al., 2021) for SSA and temperature and $h_i$ for the density. Averaging the density using a weighted average with $h_i$ preserves SWE since SWE is implicit in the weighted average equation. The average density ($\bar{\rho}$) of a snowpack with multiple layer thicknesses ($h_i$) can be

defined by:

$$\bar{\rho} = \frac{\sum_{i=1}^{n} h_i \rho_i}{\sum_{i=1}^{n} h_i} \tag{5}$$

If we replace $\sum_{i=1}^{n} h_i$ by the thickness of the whole snowpack $h_n$ and rearrange equation 5, the SWE equation is obtained:

$$\sum_{i=1}^{n} h_i \rho_i = h_n \bar{\rho} = \text{SWE} \tag{6}$$

The backscatter was estimated for the reference simulation (maximum of layering from SVS2) and 5 other grouping methods

referred to as 1-layer, 2-equal and 3-equal with the equal thickness layering method and 2-Kmeans and 3-Kmeans layers using the K-means grouping. The root mean squared error (RMSE) with the reference simulation is used to evaluate each method.

Backscatter simulations using transparent internal layers were used to estimate the influence of all the interfaces in the multi-layered snowpack configurations. The experiment referred to as transparent was done by leaving the surface to the default SMRT interface (flat Fresnel) and changing all internal interfaces to transparent which yields no reflection and full

transmission of the radar signal at each interface. The difference in backscatter was also estimated between the reference simulation and the transparent simulation. This quantifies the effect of the reflections at each snow layer interface because it is known that the backscatter in snow comes from both volume scattering and reflection at these interfaces. For example, when reducing the number of layers from 50 to 2, 48 interfaces are removed from the simulations. This reduces the overall internal layer reflections of the signal in the snowpack because of the reflected signal at each interface. However, if the permittivity

contrast between two layers is low, then the reflection will be negligible.

## 3   Results

### 3.1   Grouping and averaging methods

Figure 2 presents examples of the 2-layer equivalent snowpacks (first column) and 3-layer equivalent snowpacks (second column) derived from the two methods (equal thickness and K-means) for two different dates at WFJ. The equal thickness

method is shown by the dashed lines and the K-means method is shown by the colored symbols. The K-means method groups together layers with similar scattering, whereas the equal thickness method creates transitions that are not always consistent with transitions in scattering. For some vertical snow profiles, transitions between equal thickness layers can coincidentally occur with a change in $\kappa_e$ (for example in Figure 2) but the K-means consistently identifies changes in $\kappa_e$. One particular case of K-means grouping is seen in Figure 2f, where the classification of certain layers are mixed (groups 1 and 2). The two

layers classified as 2 at approximately 50 cm depth were "added" to the bottom layer. The effect of this particular grouping is discussed later in the section.





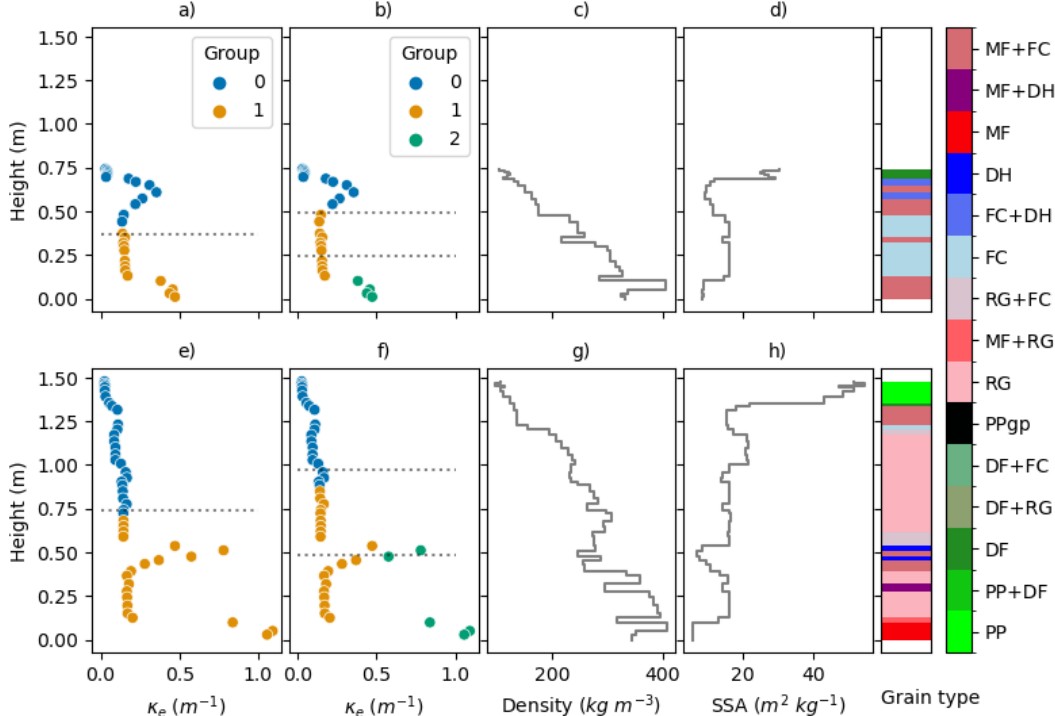

**Figure 2.** Snowpack properties (scattering, density, SSA and grain type) from Crocus and SMRT simulations at WFJ for the winter 2013-2014. The first row is 2013-12-08 and the second row is 2014-02-09. The grouping using the K-mean cluster is shown by the colors in (a,b,e,f). The $h_{norm}$ used for equal thickness grouping is also shown by the dashed lines. The colors and nomenclature for the grain types follow the international classification for seasonal snow on the ground (Fierz et al., 2009).

The grouping and averaging methods were investigated first at three different sites that represent alpine, maritime and arctic snowpacks. Figure 3 shows the simulated backscatter with SMRT for the 2013-2014 winter season. Not surprisingly, aggregating the layers into a 1-layer snowpack with averaging using the $h_i$ method (Figure 1) resulted in the highest overall

bias. Increasing the number of layers (1 to 3) reduced the bias with the reference simulation as well as using a K-means grouping with respect to equal thickness grouping. The backscatter of the arctic snowpack at TVC is well represented for most grouping methods. Using a K-means is not always better than equal grouping. For some dates, the K-means falls closely to the equal grouping (Figure 2) and the resulting backscatter is similar. Averaging using the keD method (Figure 1) was superior to averaging with $h_i$ to preserve the backscatter.

To better understand the performance of the K-means approach, Figure 4 shows biases for the 3-equal-$h_i$ and 3-Kmeans-$h_i$ layers as a function of snow properties. The 3-equal-$h_i$ method shows increased negative biases for low density, and high SSA at the WFJ and SAP sites compared to biases for the Kmeans method that were smaller and constant across density and SSA. These layers have less scattering and low SWE (low density and high SSA) with respect to other snow layers supporting the





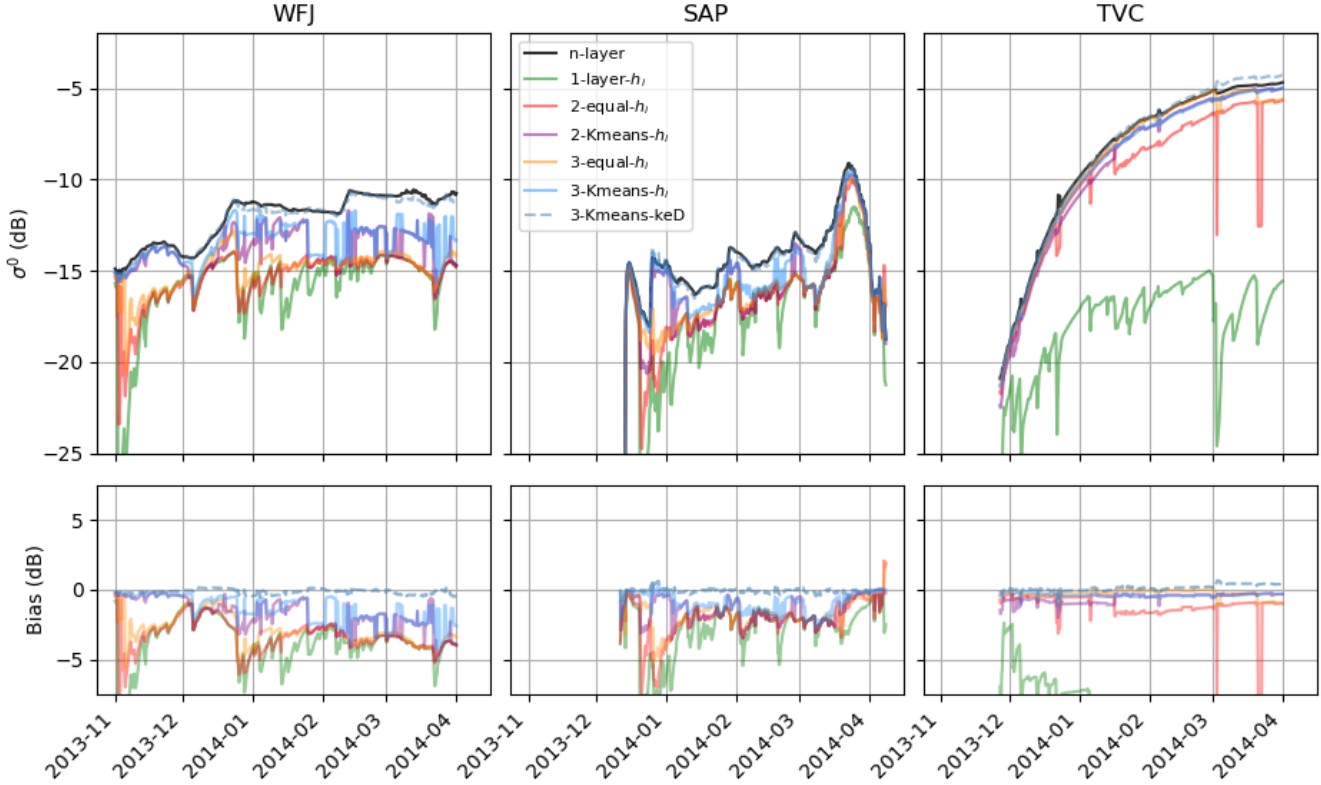

**Figure 3.** Backscatter time series for the reference simulation and different grouping and averaging methods. The backscatter (first row) and the biases (second row) with respect to the reference simulation for the season 2013-2014 are shown.

idea that those snow layers were better handled by the K-means grouping because of the ability to identify and group these transparent layers. Biases for both methods tend to increase as the snow layers were warmer which can be linked to changes in backscatter from the interface because of the contrast in the permittivity of warmer layers (liquid water). For TVC, biases remain relatively small (< 1 dB) for the majority of the season because the SWE is fairly small ( < 100 mm), and changes in stratigraphy are less frequent due to the lack of precipitation. Arctic snowpacks also tend to have a simple stratigraphy, well represented by a 2-layer snowpack (Vargel et al., 2020; Royer et al., 2021a).

Analysis of all sites and seasons (Table 2) produces results consistent with the example cases shown in Figure 3. The higher number of layers and grouping with K-means generally had better results in terms of RMSE and $R^2$ (Table 2). Again, the 3-Kmeans-keD method is the most promising method to preserve SWE and the scattering behavior because the RMSE for all sites is the lowest (RMSE = 0.5 dB and $R^2$ = 0.98). TVC, KUT and SWA had the highest RMSE (0.7-0.8 dB) compared to BYL, CDP and UMQ which had the lowest RMSE (0.3-0.4 dB). There is no pattern with respect to snow climate (alpine and arctic) and performance of the 3-Kmeans-keD method. The 2-Kmeans-keD method produced the second-best overall RMSE




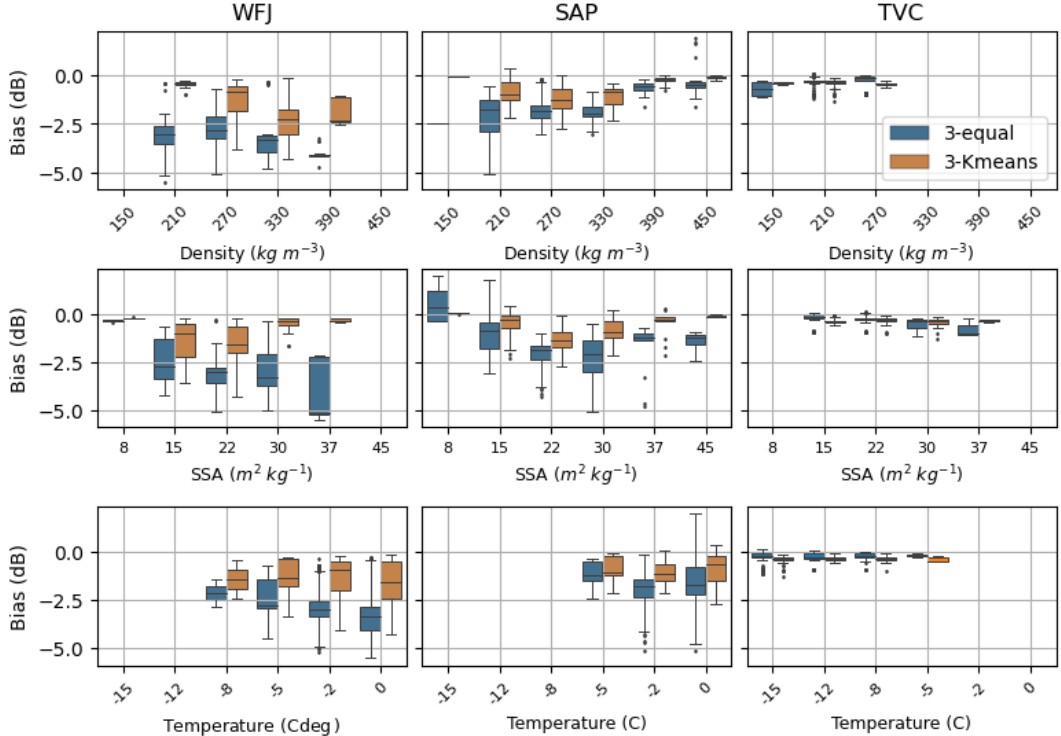

**Figure 4.** Box plot of bias in backscatter as a function of mean density, SSA and temperature per simulation for the sites WFJ, SAP and TVC for the 2013-2014 season.

of 0.7 dB and $R^2 = 0.98$ which achieved similar performance to the 3-Kmeans-keD model. This indicates that the K-means is less important in terms of preserving the microwave behavior than the keD averaging of the snow properties.

Layer averaging using a weight defined by the layer thickness is not sufficient to produce a backscatter bias under 1 dB. Averaging using $\tau = \kappa_e \, h_i$ reduced the backscatter bias under 1 dB. This method was effective in preserving the backscatter because this averaging approach puts more weight on strong scattering layers due to $\kappa_e$. The thickness is also a good indicator of scattering because a thicker layer will scatter more than a thin layer with the same scattering properties. This is an effective way to average snow properties and preserve the scattering behavior of the snowpack. The keD averaging method seems the most promising method to preserve the scattering behavior.

Transparent layer simulations yielded on average an RMSE of 0.3 dB from full layering simulations (Table 2). This indicates that on average the internal layering contributions are around 0.3 dB based on the number of layers from SVS2-Crocus simulations and the different sites and seasons. For the arctic sites (BYL, TVC and UMQ), the RMSE from the 3-Kmeans-keD method and the transparent experiment is the same, indicating that the snowpack reduction method is almost perfectly preserving the scattering behavior of the snowpack with the exception of the layer contributions. For the other sites especially



**Table 2.** Overview of the RMSE and correlation coefficient ($R^2$) of the estimated backscatter for the different grouping and averaging methods for all sites and all seasons.

| Site | 1-layer | 2-equal | 2-Kmeans keD | 3-equal | 3-equal keD | 3-Kmeans | 3-Kmeans keD | transparent |
|------|---------|---------|--------------|---------|-------------|----------|--------------|-------------|
| | | | | **Backscatter RMSE (dB), $R^2$** | | | | |
| BYL | 7.6, 0.39 | 1.8, 0.96 | 0.7, 0.99 | 1.4, 0.97 | 0.6, 0.99 | 0.6, 0.99 | **0.4, 0.99** | 0.4, 0.99 |
| CDP | 6.2, 0.64 | 3.5, 0.84 | 0.7, 0.99 | 2.5, 0.90 | 0.8, 0.98 | 0.7, 0.98 | **0.5, 0.99** | 0.3, 0.99 |
| KUT | 5.0, 0.65 | 2.5, 0.86 | 0.9, 0.96 | 1.9, 0.91 | 0.8, 0.97 | 1.1, 0.95 | **0.7, 0.97** | 0.3, 0.99 |
| RME | 4.7, 0.75 | 2.9, 0.87 | 0.7, 0.98 | 2.2, 0.91 | 0.7, 0.98 | 1.2, 0.96 | **0.5, 0.99** | 0.3, 0.99 |
| SAP | 4.4, 0.72 | 2.4, 0.88 | 0.4, 0.98 | 1.9, 0.91 | 0.5, 0.98 | 0.7, 0.98 | **0.3, 0.99** | 0.3, 0.99 |
| SNB | 2.7, 0.79 | 1.9, 0.89 | 0.7, 0.96 | 1.7, 0.92 | 0.6, 0.98 | 1.4, 0.93 | **0.5, 0.98** | 0.2, 0.99 |
| SOD | 5.5, 0.55 | 2.0, 0.91 | 0.6, 0.99 | 1.4, 0.95 | 0.6, 0.99 | 0.6, 0.99 | **0.4, 0.99** | 0.3, 0.99 |
| SWA | 4.1, 0.72 | 2.9, 0.80 | 0.9, 0.95 | 2.4, 0.84 | 0.7, 0.97 | 1.7, 0.90 | **0.7, 0.97** | 0.3, 0.99 |
| TVC | 9.0, 0.41 | 3.2, 0.71 | 0.9, 0.99 | 1.6, 0.91 | 0.7, 0.99 | 0.4, 0.99 | **0.4, 0.99** | 0.4, 0.99 |
| UMQ | 3.0, 0.78 | 2.0, 0.90 | 0.5, 0.99 | 1.7, 0.91 | 0.7, 0.98 | 0.7, 0.99 | **0.3, 0.99** | 0.2, 0.99 |
| UFJ | 4.0, 0.72 | 2.9, 0.84 | 0.9, 0.95 | 2.4, 0.88 | 0.7, 0.97 | 1.6, 0.92 | **0.7, 0.97** | 0.3, 0.99 |
| All | 5.1, 0.65 | 2.5, 0.86 | 0.7 , 0.97 | 1.9, 0.91 | 0.7, 0.98 | 1.0, 0.96 | **0.5, 0.99** | 0.3, 0.99 |

alpine sites (KUT, SWA and UFJ), the RMSE of 0.7 dB for the 3-Kmeans-keD method is larger than the layering contribution, indicating there are still some effects from the reduction method that are not accounted for.

The issue raised in Figure 2 about mixed layers between groups can be further discussed with the results of the transparent layering. The two layers at 50 cm in Figure 2f that were grouped with the layers at the bottom with 3-Kmeans do not affect the overall scattering effect of the snowpack since they will be accounted for in the weighted average using the optical thickness whether they are in group 0 or 1. The only effect was the vertical change in permittivity that was modified, creating a reduced reflection of the signal on the internal layers since the change in permittivity was less pronounced. However, the reflection of the internal layers was minimal ($\approx 0.3$ dB) so the overall effect of this special grouping case was minimal.

The simulations were performed earlier at VV polarisation since it will be the primary polarisation of TSMM but similar simulations were also done at HH-polarisation (not shown here) and yielded a higher RMSE of 0.1 dB to VV-polarisation.

## 3.2 Frequency dependence and implications for SWE retrievals

A frequency analysis was done from X-band to Ka-band (10 to 40 GHz) where volume scattering is present and includes the frequency used in this study (17.25 GHz), which is the upper frequency chosen for the Terrestrial Snow Mass Mission (Derksen et al., 2021). The errors between simulations with full layering and the 3-Kmeans-keD were similar to the values reported in Table 2 for frequency > 10 GHz, up to 30 GHz. The performance of the Kmeans-keD method is based on $\kappa_e$ which is frequency dependent. This allows us to obtain similar results when frequency (and sensitivity to volume scattering) increases. Although differences can be noted between sites, our method preserved similar performances from 10 GHz to 30 GHz. Under the 10





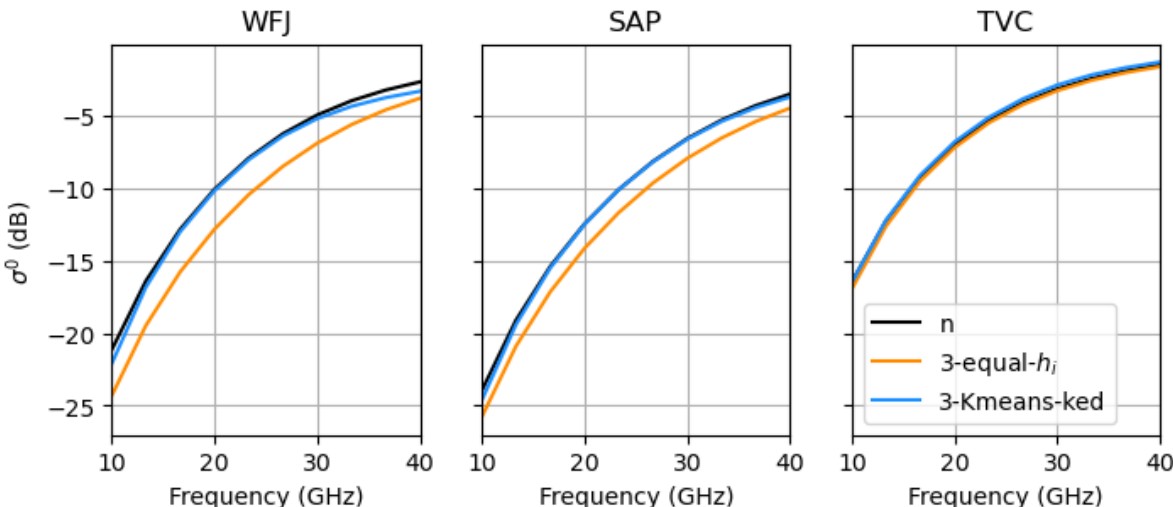

**Figure 5.** Backscatter simulations of 3-equal-$h_i$ and 3-Kmeans-keD methods as a function of frequency

GHz, a different simulation setup would be needed since the soil contribution will dominate. For frequencies > 40 Ghz, the method based on $\kappa_e$ became similar to the 3-equal-$h_i$ because the backscatter comes from the snow surface as the frequency increases and the penetration depth decreases.

### 3.3 Computation efficiency

Computation time was also estimated for each method to evaluate the gain in computational efficiency. The reduction in computation time of the backscatter for reducing a complex multilayered snowpack (50 to 10 layers) to 2 or 3 layers is shown in Table 3. The largest reduction in computation time was from 50 layers to 2 layers with 87% and the smallest was from 10 to 3 layers with 39%. Even the smallest reduction is considerable and motivates this work in the context of operational SWE retrieval implementations, where these computations have to be done at large scales.

## 245 4 Conclusions

In this paper, we showed the performances of different methods to simplify complex multilayered snowpacks to less than or equal to 3 layers while preserving their microwave scattering behavior and bulk snow mass. We evaluated our method using simulated snow properties generated by the Crocus snowpack scheme at 11 sites which were input to the SMRT model to calculate backscatter at 17.25 GHz and VV polarization. This emulates potential future measurements from the Canadian
Terrestrial Snow Mass Mission. The method was a K-means clustering algorithm that groups the snow layers based on the extinction coefficient and the height of the layer in the snowpack. Then, a weighted average using the extinction coefficient and the thickness was applied to the snow properties except for density for which snow layer thickness was used as a weight





**Table 3.** Computation time of backscatter with SMRT and grouping methods. Results in various configurations of the number of layers are shown here. The grouping times for the Kmeans method are also shown.

| Layering | $\sigma^0$ computation time | Reduction in $\sigma^0$ computation time | |
| --- | --- | --- | --- |
| | | to 3 layers | to 2 layers |
| 50 layers | 0.90 s | 83 % | 87 % |
| 40 layers | 0.80 s | 81 % | 85 % |
| 30 layers | 0.67 s | 78 % | 82 % |
| 20 layers | 0.46 s | 68 % | 75 % |
| 10 layers | 0.25 s | 39 % | 53 % |
| **Grouping** | $\sigma^0$ **computation time** | **Grouping computation time** | |
| 3-Kmeans | 0.15 s | 0.04 s | |
| 2-Kmeans | 0.12 s | 0.03 s | |

to conserve SWE. Grouping the original 50 snow layers into 3 layers using this method reproduced the snowpack backscatter of the original multi-layered snowpacks with an overall RMSE = 0.5 dB and $R^2$ = 0.98. Using this methodology in SWE retrieval algorithms allows snowpack simplification without impacting the scattering behavior or compromising the geophysical properties. Reduction in mathematical complexity of SWE cost function and reduction in computation up to 80% can be gained by using fewer layers in SWE retrievals.

We proposed a simple method that can also be applied to other fields of study that need to simplify a multilayered snowpack without compromising on the electromagnetic properties of snow. For the TSMM workflow, this algorithm allows to simplify the modeled snowpack from the SVS2-Crocus to the SWE retrieval. It can also be used in the assimilation schemes to link the posterior or "optimized" snowpack produced by the SWE retrieval. For these reasons, this method offers an effective way to link physical snow modeling and snow radiative transfer modeling together in SWE retrievals.

*Code and data availability.* Code are Available on at https://github.com/JulienMeloche/radar_equivalent_snowpack

*Author contributions.* JM, NL and BM wrote the manuscript with contributions from all co-authors. All co-authors designed the experiment. JM and NL performed the analysis. VV and NL developped SVS-2. All co-authors reviewed the manuscript and provided analysis guidance.

*Competing interests.* Some authors are members of the editorial board of The Cryosphere.



*Acknowledgements.* The study was supported by Environment and Climate Change Canada and made possible by open source developpement of the SMRT and Crocus model.

# Appendix A

**Table A1.** Crocus schemes and parameters used at the different sites. The meaning of the Crocus schemes can be found in Lafaysse et al. (2017) and Woolley et al. (2024). The different options are defined in their respective paper; B21: modified C13 from Carmagnola et al. (2014), B92: Brun et al. (1992), R21: Royer et al. (2021b), V12: Vionnet et al. (2012) and Y81: Yen (1981). The $\gamma$ parameter represents the snow ageing coefficient and was determined at the sites in (Gaillard et al., 2024), except for Reynolds Mountain for which the default value of 60 days was used.

| Site | Metamorphism | Radiation | Snowfall density | Thermal conductivity | Water percolation | Compaction | $\gamma$ (d) |
|---|---|---|---|---|---|---|---|
| Bylot | B21 | B92 | R21 | Y81 | B92 | R21 | 900 |
| Col de Porte | B21 | B92 | V12 | Y81 | B92 | B92 | 20 |
| Kühtai | B21 | B92 | V12 | Y81 | B92 | B92 | 10 |
| Reynolds Mountain | B21 | B92 | V12 | Y81 | B92 | B92 | 60 |
| Sapporo | B21 | B92 | V12 | Y81 | B92 | B92 | 40 |
| Senator Beck | B21 | B92 | V12 | Y81 | B92 | B92 | 60 |
| Sodankylä | B21 | B92 | R21 | Y81 | B92 | R21 | 150 |
| Swamp Angel | B21 | B92 | V12 | Y81 | B92 | B92 | 60 |
| Trail Valley Creek | B21 | B92 | R21 | Y81 | B92 | R21 | 900 |
| Umiujaq | B21 | B92 | R21 | Y81 | B92 | R21 | 200 |
| Weissfluhjoch | B21 | B92 | V12 | Y81 | B92 | B92 | 200 |



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
