# Peer review of "Radar Equivalent Snowpack: reducing the number of snow layers while retaining its microwave properties and bulk snow mass"

_EGUsphere, 2024_

## Author Comment (AC1)

**Radar Equivalent Snowpack: reducing the number of snow layers while retaining its microwave properties and bulk snow mass**

**Note to editor:**
We thank all the reviewers for helpful, constructive and thorough comments that helped improve the manuscript. All comments were considered and are addressed in the document below.

Reviewer's comments (R1, …)
Answers to reviewer.
*Modification to text. They are visible(yellow) in the track change version.*

**Reviewer: 1**

Comments to the Author

1) Lines 35-38, end of paragraph "…, but can prove challenging for remote sensing applications." Could you elaborate on some of these challenges?

This sentence was removed from the paragraph.

2) Your proposed method is based on K-means clustering. Being a central part of the proposed method, could you add more detailed description of the method? (sections 2.4. and 3.1. of the manuscript).

We agreed that some details are missing on the K-means algorithm. This part was added at line 144.

*K-means clustering (Ikotun et al., 2023) identifies groups within the parameter space by minimizing the variance within each group or cluster. First, it randomly initializes centroids for each group in the parameter space and then assigns each point to the initial groups based on the Euclidean distance to the nearest centroid. The centroids are updated to the mean position of all points within each group. The process is repeated iteratively until a convergence is reached (when the centroid positions no longer change significantly) or a fixed number of iterations is completed.*

3) The K-means clustering algorithm is known to converge to a local minimum, which is not necessarily a global one. Therefore, the results can depend on the chosen initial points of the cluster means. The initial points should be mentioned in the manuscript.

We agree with your comment. We used the K-means++ initial method to choose our initial centroids and avoid this known problem of the K-means algorithm. This is the default option in the scikit learn package from scipy, which was used for this study. This part was added in section 2.4.

*A known issue with K-means is that the random initialization of the centroids can lead to non-representative clusters due to a local minimum reached in the convergence. To avoid this issue, the K-means++ initialization (Arthur and Vassilvitskii, 2007) was used which ensures a smart initial choice of the centroids based on the empirical distance distribution of the points, essentially selecting centroids that are the furthest from each other. This speeds up the convergence and improves the quality of the clusters.*

4) It would be interesting to see an example of the plane with the data points, initial mean points, and the converged clusters. If you don't consider this figure informative, it does not have to appear in the manuscript, but perhaps you could produce it in the comments?

Because we now specified the use of the K-means++, we don't see the benefit of adding this in the paper. However, below is a figure of an example of the initial cluster computed the K-means++ method and final cluster. The x markers represent the initial K-means++ centroids, and the circle represents the final centroids.

[Figure]

5) You are using a two-dimensional parameter space in the K-means clustering. Could you comment on the choice of the parameter space? Is there a reason to consider the extinction coefficient instead of both scattering and absorption coefficients in a three-dimensional parameter space?

That is a fair point. In previous work, we tried to only use the scattering coefficient but the best results came with the extinction coefficient. Future work will focus on augmenting the parameter space to improve this method if needed.

6) In section 2.4. of the manuscript, you describe the K-means clustering and the equal thickness grouping in rather equal terms, although the proposed method uses the K-means clustering. If you find it appropriate, please consider reorganising the section so that the K-means is described first as the primary method, and the equal thickness grouping is described then as a secondary method that is used for comparison. In my opinion, this would make the section easier to read.

Agreed, we made the modification in the revised manuscript, section 2.4. The K-means method is first introduced followed by the equal thickness grouping method.

7) In section 2.4. of the manuscript, you describe the weighting of the more complex snowpack into the simplified layers. You use the thickness based weighting (h-weighting) for the equal thickness grouping and optical thickness based weighting (τ-weighting) for the 3- and 2-means clusters. Does this not make the comparison between the two clustering methods unfair?

The optical thickness-based weighting was also used with the equal thickness grouping method as shown in Table 2. This was done to estimate the effect of both the clustering and the optical thickness average. We modified the text in section 3.2 (line 206):

*The 2-cluster with $\tau$-average and 3-equal with $\tau$-average method produced the second-best overall RMSE of 0.7 dB and $R^2$ = 0.97. The 3-equal with $\tau$-average method achieved similar performance to the 3-cluster with $\tau$-average. This indicates that the K-means cluster is less important in terms of preserving the microwave behavior than the $\tau$-average of the snow properties.*

We also added a sentence in section 2.4 to clarify:
*Both averaging methods were tested on the equal and cluster grouping to compare the performance of each method.*

8) The last paragraph of section 2.4. (lines 162-170) describes the problem of removing interfaces when merging layers. This is a very important and an interesting point. However, I found the paragraph somewhat difficult to read. If you agree, please consider reorganising the paragraph so that the problem is introduced first, and then the solution.

This is a good point. The paragraph was reorganized, see revised manuscript.

*It is known that the backscatter in snow comes from both volume scattering and reflection at these interfaces. although interface reflection is small with respect to volume (except at nadir). When reducing the number of layers from 50 to 2, 48 interfaces are removed from the simulations. Although interface reflection is small with respect to volume, this reduces the overall internal layer reflections of the signal in the snowpack because of the reflected signal at each interface. However, if the permittivity contrast between two layers is low, then the reflection will be negligible. To quantify the effect of the reflections at each snow layer interface, backscatter simulations using transparent internal layers were used to estimate the influence of all the interfaces in the multi-layered snowpack configurations. The experiment referred to as transparent was done by leaving the surface to the default SMRT interface (flat Fresnel) and changing all internal interfaces to transparent which yields no reflection and full transmission of the radar signal at each interface. This means setting transmission to 1 and reflection to 0. The snow-ground interface was not modified. The difference in backscatter was estimated between the reference simulation and the transparent simulation to estimate the contribution of the n-2 internal interfaces reflection that are removed when reducing the number of layers.*

9) Paragraph on lines 221-226 discusses the case when K-means finds clusters of layers that are not connected. In the averaging of such layers, scattering and absorption properties of the layers are not the only things affecting, but also the (optical) depth of the layers in the snowpack. For example, if two layers with equal thickness and extinction coefficient are placed at the top and bottom of the snowpack respectively, their effect to the total backscatter is not the same. In such a case moving one of the layers either from top to bottom or from bottom to up might not make sense. Could you comment on this?

We agree with this comment since the attenuation is not the same for a buried layer than for surface layer. However, in our case, this extreme case is unlikely to happen since the layers would be pretty far from each other in the parameter space. Here, it is really when some layers are both close in terms of scattering but are not "direct neighbors" in the snowpack as illustrated in Figure 2. Restricting only neighboring layers in the K-means would restrict much of the clustering and was a methodological decision we made. In the end, we can see from the results that this effect is probably present but less than 1 dB for all sites. We added a sentence on this issue and modified the following

sentence.

*However, attenuation of the scattering from these layers can differ if the layers are moved upward or downward in the snowpack. The other effect was the vertical change in permittivity that was modified, impacting the reflection of the signal on the internal layers. However, because the reflection of the internal layers was minimal ($\approx 0.3$ dB) and change in backscatter was < 1 dB for all sites, it was concluded that the overall effect of this special grouping case was minimal.*

**10) The proposed method is based on an idea of running a snow process model with a high number of layers. However, this can also be computationally challenging. Can you comment on how computationally demanding it is to run the snow process model compared to the radiative transfer model? How does the computational cost increase with increasing number of layers (e.g. is running a 4-layer simulation much less expensive than running a 40-layer simulation?)**
Running our snow process model at multiple layers is not much more expensive than at 3 layers. For example, running the model with 3 layers comes at an average of 3.0 s per season compared to an average of 3.7 s per season for a 50-layer simulation. This is substantially less than with radiative transfer.

**11) Your proposed method merges the snow layers based on their microwave properties, whereas the snow process model merges them (when the maximum number of layers is reached) based on their physical properties. The latter preserves the physical properties of the snowpack, while in the former does not. In terms of the computational cost of the radiative transfer simulation and the complexity of the cost function, the same benefits are obtained by running a snow process model with the maximum number of layers set to three. How does the simulated backscatter from such a snow process model run compare with that simulated using your proposed method? Does your method have a clear advantage in terms of the backscatter when both options are compared with the 50-layer run?**
Running the snow model with a few layers (e.g. 3 layers) does not preserve the physical properties compared to running with > 10 layers, it would preserve the mass. The simulation of layered snow microstructure depends in part on simulated temperature profiles, which are not well represented when using a few model layers. Therefore, running our snow model with fewer than 10 layers results in poorly represented vertical temperature profiles, hence, layer microstructure. The introduction was modified to clarify this:

*Numerical snow models that use large numbers of layers can improve the representation of the dynamic physical processes within the snowpack, such as heat and mass fluxes, resulting in a better representation of the temperature profile. A better simulation of the vertical temperature profile within the snow improves the simulation of microstructure evolution and spring snowmelt initiation (Cristea et al., 2022) or the identification of weak layers in the context of avalanche hazard forecasting (Morin et al., 2020). Therefore, there is a benefit to adding layers in physical modeling and improve the full vertical profile of snow properties.*

Below is an example of Crocus simulations over 1 season (2013-2014) at WFJ. We show the SWE for Crocus simulations with 3, 5, 10, 20, and 50 layers do not change but that the backscatters vary wildly between the simulations but become more similar to each other for simulations with > 10 layers

12) How could your proposed method be applied to different snow retrieval problems?

- Is it only suitable for retrieving SWE or can it also be used to retrieve other snow variables? When simplifying the snowpack, the averaging of the physical snow variables is done using extinction coefficient. Therefore, the effective values don't have the same physical meaning. Does this limit the potential application of the proposed method?

Yes, it is mostly suitable for volume scattering SWE based retrieval, but it can also be used in other microwave retrieval problems. The algorithm aims at focusing on the layers that are relevant based on a given frequency. The snow properties values have an electromagnetic physical meaning, so the potential application must be specific to this. This was added to the conclusion.

*The algorithm averages snow layers to obtain effective layers with snow properties that have an electromagnetic equivalent, so the potential application must be specific to the chosen frequency. This method could be adapted to passive microwave remote sensing where the signal is also highly dependent on snow scattering, including ice sheet or sea ice remote sensing where the microwave signal could be simplified to focus on radiative relevant layers.*

- In the Bayesian retrieval, the cost function is defined as the sum of the mismatch term (between the observed and the modelled microwave signature) and the prior term. Your proposed method affects both terms; the mismatch term through the forward model configuration (how many layers are assumed) and the prior term. Can you comment on the use of the proposed method in this case?

We added these two sentences in the conclusion.

*In the TSMM Bayesian retrieval, the mismatch term is affected by running the radiative transfer solution and optimizing the posterior (parameters) to a reduced number of layers. The prior*

*distribution on snow properties from SVS2 are obtained with the algorithm again to a reduced number of layers.*

- On the other hand, in the data assimilation approach (e.g. particle filter), your method seems to have a more straightforward application to simplify the snowpack before the radiative transfer simulation to save computation time in the radiative transfer simulation. However, this approach would require an ensemble of snow process model runs, which can be computationally expensive. Can you comment on the use of the proposed method in this case?

Yes, the method would be used to reduce computation in a particle filter by simplifying the calculation of backscatter for all the ensemble snow members. The backscatter would be calculated on a reduced number of layers for each snow member. This was added in the conclusion:

*It can also be used in assimilation schemes to reduce the computational time required to calculate a backscatter ensemble from a collected of snowpack members.*

Reviewer 1 technical comments:

T1) Line 151: "We investigated three different ways of averageing: …". Only two are mentioned in the text.
Text was modified to "two different ways".

T2) Line 26-27: "More typically, observations are unavailable, so snowpack information must come from …". "must" is perhaps too strong word?
 Text was changed to "can come from"

T3) Line 104: "The choice of the electromagnetic model does not influence the final result of this method, …". This could be wrongly understood to mean that the used electromagnetic model does affect the simulated backscatter (the result) and through that, the conclusions of the manuscript. Could you use another expression?
This sentence was removed to avoid any confusion.

T4) Line 119: "where $p\_11=$ … and $p\_22 =$ … are defined from the phase function." repeats line 116.
The first definition (line 116) was removed.

T5) Line 121: Reference should be (Sihvola,1999)?
The reference was corrected in the text.

T6) Lines 145,149,150,etc.: I don't think symbols after text require parenthesis. E.g. "extinction coefficient ($\kappa_e$)" --> "extinction coefficient $\kappa_e$" and "layer thickness ($h_i$)"  --> "layer thickness $h_i$".
The parentheses were removed

T7) Should section 3 of the manuscript be "Results and Discussions"?
Yes, the section was modified

T8) Lines 176,177: The word "scattering" is used but is supposed to be "extinction"?
Yes, the word was changed to "extinction".

T9) Suggestion: Consider using "h-weighting" and "$\tau$-weighting" (instead of keD) for the two layer-averaging methods.

Thanks, we modified as you suggested.

T10) Line 190-191: "3-Kmeans-hi" can be slightly confusing. "K" stands for the number of clusters; would it make more sense to write "3-means"? Also, in my opinion separating the clustering and averaging methods could make the expression clearer. Suggestion: "3-means clustering with h-weighting." Apply as you see fit.
Thanks, we modified as you suggested.

T11) Similarly, "3-equal-$h_i$" could be "3-equal-thickness" or "3-equal-h" omitting the subindex i as it is used to refer to the 50-layer snowpack. Using "3-equal-thickness" for clustering and "h-weighting" for the weighting could help to avoid confusion between the two. Apply as you see fit.
Thanks, we modified as you suggested.

T12) Lines 202,205: Same suggestion; consider changing "3-Kmeans-keD" to something along the lines "3-means clustering with τ-weighting" or "3-means clustering with τ-averaging". Apply as you see fit.
Thanks, we modified as you suggested.

T13) Figure 3: Y label of the bottom figures should be "difference" as "bias" represents systematic or average difference. Same in figure caption.
Bias was changed to difference

T14) Line 230: "Frequency analysis" has other meanings. Consider different expression.
It was changed to "An investigation on the frequency dependence"

**Reviewer: 2**

1) One question for this topic is whether the layer simplification was utilized directly in the snow process simulation (e.g., using Crocus) or it was solely utilized in the microwave simulation. A clarification may be needed in the abs.
The simplification was only done on the snow properties prior to the microwave simulation and after the Crocus simulation. We modified the abstract by adding the following sentence:

*The layer simplification is done as an intermediate step between the physical modeling (SVS2-Crocus) and the microwave radiative transfer (SMRT)*

2) Another point is that the authors did experiments for 11 sites globally, whereas only results from three sites were explicitly presented in Fig. 3-5. However, I may be more interested in knowing how snow differs in different sites and what is the key characteristic (wind slab, melting?) that results in the requirement for 2-3 snow layers.

The three sites were selected because they represent most snowpack conditions (Arctic, dry and wet alpine). However, some details are missing as you mention. We added this part to section 2.1 .

*From the 11 sites, 3 were selected to more easily illustrate the methodology. These sites (TVC, WFJ and SAP) have distinct snowpack characteristics. The TVC arctic snowpack is characterized by a layer of highly scattering depth hoar with a dense wind slab on top. The WFJ alpine snowpack is characterized by a deep snowpack with progressive density increase from top to bottom with some melt-freeze crusts due to warming events throughout the season. The SAP snowpack is characterized by a similar alpine snowpack but more impacted by wet precipitation.*

3) The authors didn't clearly state whether they focus mainly on dry snow.

We only focused on dry snow in the context of SWE retrievals for a volume scattering method in mind. We added this in the introduction.

*This study focuses only on evaluating our snowpack reduction method on dry snow in the context of SWE retrievals based on volume scattering. However, this method could potentially be used for wet snow since the extinction coefficient would be sensitive to liquid water via the absorption coefficient if the melt is correctly estimated by the physical model.*

4) Lines 35-38: Actually, multiple layers provide a continuous, high-resolution temperature profile inside the snowpack, which benefits most to the accuracy of snow microstructure simulation. To retain snow mass, people don't need a fine-resolution snow process model. In addition, this kind of model has great potential to accurately simulate the snowmelt process, because the fine-resolution snow layers can have different snow temperatures, for the model to evaluate whether the melting point has met.

Thanks, we added your point about the microstructure and snowmelt.

*Numerical snow models that use large numbers of layers can improve the representation of the dynamic physical processes within the snowpack, such as heat and mass fluxes, resulting in a better representation of the temperature profile. A better simulation of the vertical temperature profile within the snow improves the simulation of microstructure evolution and spring snowmelt initiation (Cristea et al., 2022) or the identification of weak layers in the context of avalanche hazard forecasting (Morin et al., 2020).*

5) Line 40: In Pan et al. (2017), only two snow layers were retrieved. Therefore, it does not have the "large numbers of layers" problem mentioned in the lines 42–43 followed. This part needs a revision.

We agree, this is not what we meant. We revised the whole paragraph, see the revised manuscript.

*Some algorithms couple a physical snow model and a snow RTM to retrieve SWE using microwave remote sensing data (Langlois et al., 2012; Larue et al., 2018; Singh et al., 2024). Snow RTMs can model the radar backscatter using snow parameters from complex layered snowpacks. In a SWE retrieval like Pan et al. (2017), the SWE (depth and density) of the different layers is estimated by minimizing the difference between the modeled and measured backscatter. To simulate the backscatter, most RTMs solve the radiative transfer equation based on the discrete ordinate and eigenvalue method (Picard et al., 2004), which discretizes the radiative transfer equation and solves a nonhomogenous system of linear equations based on the number of layers. Increasing the number of snow layers thus increases the computational cost at many levels within the retrieval algorithm. Also, a larger numbers of layers increase the complexity of the retrieval by increasing the number of variables in the cost function. This is why current retrievals typically use a two-layer model (Saberi et al., 2021; Pan et al., 2017). Completely neglecting stratigraphy by using a one layer model can affect the performance of the retrieval (Durand et al., 2011) because layering strongly influences the backscattering properties of snow (Rutter et al., 2016). A one-layer model oversimplifies the scattering behavior of the snowpack and so is not adequate in most cases (Rutter et al., 2019; Meloche et al., 2024; Montpetit et al., 2024). For this reason, a two or three-layer model provides notably better SWE retrievals by accounting for stratigraphy in a certain way (Pan et al., 2017; Saberi et al., 2021). In the end, there is a disconnect between needing several layers in physical model to simulate a realistic microstructure profile but needing only 2 or 3 layers in SWE retrievals for computation simplicity.*

6) Lines 44-49: It is suggested to separate the retrieval studies that considered only one layer and

two layers. The use of two layers had been a tradeoff between simulation accuracy and retrieval feasibility. It is ok that the authors think they should at least consider one more layer, i.e., to consider 3 layers in total. However, the following sentence sounds really strange in the current manuscript:

"This is why current retrievals typically employ a one or two-layer model (Saberi et al., 2021; Durand et al., 2024; Pan et al., 2017). However, neglecting stratigraphy by using a small number of layers model reduces the performance of the retrieval (Durand et al., 2011) because layering strongly influences the backscattering properties of snow (Rutter et al., 2016)."

Actually, the studies in Saberi et al., 2021 and Pan et al., 2017 have considered the snow stratigraphy in some sense.

We agree, this is not what we meant. The message is that we need to consider at least 2 layers for SWE retrievals and there is a disconnect with Crocus since it needs a lot of layers to properly simulate a realistic microstructure profile. We revised the whole paragraph, see the revised manuscript and previous comment.

7) It is suggested to consider this reference:

Yu, Y., Pan, J., Shi, J., 2021. Evaluation of the effective microstructure parameter of the microwave emission model of layered snowpack for multiple-layer snow. Remote Sens. 13. https://doi.org/10.3390/rs13102012.

By retaining air-snow and snow-soil boundary reflectivities using the topmost and bottom-most snow density instead of the profile-average density, the idea of Yu et al. (2021) becomes an indirect supporter of your 3-layer configuration suggestion.

Thank you for suggesting this paper we didn't know about. We added this in the introduction.

*Yu et al. (2021) proposed an interesting method to estimate an effective 1-layer snowpack for passive microwave applications that calculates a SWE weighted average for the microstructure parameter and preserves the reflectivity of the air/snow and snow/ground interface from the multilayer snowpack. With this approach, the scattering properties are better preserved. To our knowledge, a robust method still does not exist to effectively reduce the number of layers of a given snowpack while minimizing changes in scattering properties.*

8) Line 56: The snow type in Pan et al. (2017) was taiga snow. It does not have a wind-slab layer.
True, the reference was removed.

9) Line 5: According to line 65, it is better to say, "2-3 layers" directly in the abs.
We modified the abstract to 2-3 layers

10) Line 88: Could you add more details for the wind-slab and basal vegetation consideration and state why they are important in your study?
We added this sentence.

*This allows for a better "arctic" density profile by increasing the wind slab density and lowering the depth hoar density.*

11) Line 127: The use of "presented" is confusing here. Better say, could be different for different snow grain types, according to Picard et al., 2022.
We removed the last part of the sentence to avoid confusion.

*The polydispersity was assumed to be 1 for all grain types in this experiment but future work could*

*include a polydispersity for different Crocus-simulated grain types.*

12) Line 130: However, the key point and importance for evaluating using different frequencies is that one can use a single profile to simulate multiple frequencies correctly, despite different penetration depths.
This is a fair point. We would need to investigate this in another study but since TSMM frequencies are fairly close to each other, this not something we anticipate being a problem.

13) Section 2.4: A K-means classification will not work well directly, because you usually will not be allowed to combine layers that are not adjacent. More details are needed here.
We added more details on the K-means classification, see comment 2 from reviewer 1.

14) Line 153: To preserve mass and snow depth, one can also actually only tune the snow microstructure parameter, because any change you make will not go into the subsequent CROCUS simulation. The snow layer simplification for RT calculation was operated stand-alone.
We believe that also including the temperature is important because it impacts the permittivity and the absorption coefficient. It also makes the methodology compatible to passive microwave remote sensing because the physical temperature is important in the brightness temperature calculation.

15) Figure 2 shows two examples where the existence of surface wind slabs and intermediate melt-freeze crusts inside the snowpacks can be two reasons to suggest using 3 layers, respectively. Are the first and second rows for the same site?
Yes they are from the same site but different time in the season.

16) Line 189: Which does the "keD method" refer to in the context of Section 2.4?
Following the comment from reviewer 1, this notation was removed.

17) From Figure 2, did you compare the differences in efficiency between applying K-means grouping on extinction coefficients and on original snow physical parameters (SSA, density, etc.)?

Yes, we tried this, and separating the scattering and absorption coefficient and the extinction coefficient was always better.

18) Lines 195-196: The error increases with increasing snow depth and increasing SSA, too, as the snowpack evolves with time. From my experience, your results shown in Fig. 3 have not included the snowmelt period yet. Therefore, it should not be caused by warmer snow temperatures, at least solely. By the way, it could be better to add subplots showing snow depth and air temperature time series together in Fig. 3.
We agree with this comment and we removed that sentence. This is probably due to the grain size error and not caused by warmer temperature.

19) Lines 214: What does "transparent layer simulation" mean again? Did you mean mandatorily setting the interface transmissivities to 1 and interface reflectivities to 0, although there are 3 layers with different refraction indexes?

Yes, the transparent layer simulation does mean setting the interface transmission to 1 and reflectivity to 0 but the number of layers did not change. This is to estimate the backscatter contribution of the $n - 2$ interfaces in the simulation. We added this in the methodology:

*This means setting transmission to 1 and reflection to 0. The snow-ground interface was not modified. The difference in backscatter was estimated between the reference simulation and the transparent simulation to estimate the contribution of the n-2 internal interface reflections that are removed*

*when reducing the number of layers.*

---

## Author Response (AR2)

Manuscript egusphere-2024-3169 version 2

**Radar Equivalent Snowpack: reducing the number of snow layers while retaining its microwave properties and bulk snow mass**

**Note to editor:**
We thank all the reviewers for helpful, constructive and thorough comments that helped improve the manuscript. All comments were considered and are addressed in the document below.

Reviewer's comments (R1, …)
Answers to reviewer.
*Modification to text. They are visible in green in the track change version.*

**Reviewer: 1**

General comments
• The core idea in the paper, the application of the K-clustering method to reduce the complexity of the snowpack while retaining the backscattering properties and snow mass, is nice and straightforward. However, compared to this, the paper feels a bit cluttered. I think the manuscript would benefit from more clarity.
• For example, are all the 6 reference methods needed? I recognize that together they show how much things can go wrong if one considers the simplest possible case (single layer snowpack), but on the other hand, is this essential to the main point of the manuscript? Is this additional information more important than the distraction that it adds to the results?
• In some instances, the notation is a bit confusing (see specific comments for examples).
• I hope that the authors would check the manuscript again and look for places to improve the clarity beyond the specific comments listed below.

We tried to improve the clarity of the paper by integrating all specific comments and some other improvements directly into the manuscript. We improved the consistency of the terminology in the method and removed some reference methods in Figure 3. Section on computation efficiency was moved to the appendix and a small section on SWE retrievals implication was added.

Specific comments
1. Lines 3-4: "…, with more complex layering yielding richer information but at increased computational cost." It should mention also the increased number of unknowns, since the context is retrieval process.
Thanks, we added "number of unknowns" at the end of the sentence.

2. Line 5: "…, while preserving the microwave scattering …" Perhaps "preserving" is too strong term here, since the scattering does change in the process.
It was changed to "nearly preserving".

3. Line 7: "Then, a weighted average…" This sentence seems disconnected from the previous sentence.
The text was modified to bring to the two sentence together.

4. Lines 13-14: "SWE retrievals can be applied to simplified snowpacks, while maintaining similar scattering behavior without compromising the modeled snowpack properties": The use of the method for retrievals was not demonstrated in the paper.

Indeed we are not showing the use of the method in a retrieval, we are simply mentioning that this method can be used in SWE retrieval applications. This sentence was changed to: "In SWE retrieval applications, this method can be used to simplify snowpacks and reduce the number of variables to optimize, while maintaining similar scattering behavior without compromising the modeled snowpack properties."

**5. Lines 39-40: "… or the identification of weak layers in the context of …" is this part relevant?**
This part was removed.

**6. Line 44: "SWE (depth and density)" -> "SWE (function of depth and density)"**
"Function of" was added

**7. Lines 69-71: Sentence starting with "If a method that …" sounds bit weird, please check.**
The sentence was modified as follows:
*This would allow the accuracy of SWE retrieval not to be compromised and the computation time to be reduced.*

**8. Lines 119-142 contain quite specific description of the RT modeling. Is this necessary for this paper?**
We think it is necessary since it shows how the extinction coefficient is calculated which is central to the methods described in paper.

**9. Line 153-154: Ikotun et al. is referred twice.**
The second citation was removed, and to avoid repetition, we changed the beginning of the sentence to:
*This algorithm identifies groups …*

**10. Line 163: Unnecessary parenthesis, (\kappa_e) -> \kappa_e, please fix elsewhere also.**
Parenthesis were removed.

**11. Line 166: "(referred to equal thickness)" -> to as**
"as" was added

**12. Lines 165-170: h_norm is not defined here? Also, the paragraph discussion h_norm seems to cut between two parts of the text that both discuss K-means clustering. Please consider clarifying and restructuring.**
We added this to define h_norm.
*… based on the normalized height h_norm which is the height of a layer divided by the thickness of the whole snowpack.*

We also rearrange the whole paragraph. This part was moved earlier in the paragraph.
*The grouping of layers was done in two ways. The first method was used as a baseline comparison. This method (referred to as equal) aggregated layers based on the normalized height h_norm which is the height of a layer divided by the thickness of the whole snowpack. For instance, two equal thickness snowpack, the top half of the snowpack, i.e all the layers with h_norm greater than 1/2 of the snowpack thickness would be aggregated into a first layer and the bottom half would contain all the layers with h_norm that are less than or equal to ½ of the snowpack thickness. This method is a basic way to obtain an equivalent snowpack with layers of equal thickness. Also, a 1-layer simplified snowpack was considered by grouping all the layers into 1 group to evaluate the worst-case scenario in reducing the number of layers.*

**13. Also, this paragraph could be otherwise improved. E.g. the sentence "The 1-layer was created by**

grouping all layers into 1 group" is a bit weird.
The sentence was modified.

*Also, a 1-layer simplified snowpack was considered by grouping all the layers into 1 group to evaluate the worst-case scenario in reducing the number of layers.*

14. Lines 171-173: Use of h_group is a bit confusing. Is "group" used here as an index, i.e. group = 1,2,3? On the other hand, is the symbol used anywhere in the text after (is it necessary)?
The symbol was removed.

15. Line 174: "(referred to as h_i)" -> "(… h_i-average/averaging)"
The whole paragraph was revised. Link to previous comment.

*Once the grouping of layers was done, the snowpack layer properties (density, temperature, and SSA) were averaged. We investigated two different ways of averaging: (1) a weighted average based on h as a baseline (referred to as the h-average), and (2) a new weighted average based on the optical thickness τ=ke×h (Zhu et al., 2021) for SSA and temperature (referred to as the τ-average). The h-average is used for density in both average methods, as it ensures the conservation of SWE. The average density of a snowpack with multiple layers of thickness hi can be defined by:*

16. Line 179: "… whole snowpack h_n …": Confusing to use n as a subindex, how does it differ from h_i?. E.g. h_snow could be better.
We change the index to h_snow.

17. Schematics in Figure 1 could be improved to clarify the process. At the moment it is difficult to interpret. I think what should be highlighted is the fact that the process in this paper has two levels, the grouping method and the averaging method, both of which have multiple options, and the different combinations of these two produce the "methods" that are being compared. In addition, some technical points: 0.33 and 0.66 or 1/3 and 2/3? The word "group" appears both as italic and non-italic text. Also, please make the equations less vague, e.g. h_group = \sum h_i, what is summer over?
We added this sentence before Figure 1.

*The method is divided into two operations applied on layers: grouping and averaging.*

We made the suggested modifications to Figure 1.

[Figure]

**Multi-layered snowpack**
Snowpack
50 layers x 4 properties

**Grouping**

**Equal**

**1 layer:** group 1
**2 layer:**
$$f(x) = \begin{cases} group\ 1, & h_{norm} \leq 1/2 \\ group\ 2, & h_{norm} > 1/2 \end{cases}$$
**3 layer:**
$$f(x) = \begin{cases} group\ 1, & h_{norm} < 1/3 \\ group\ 2, & 1/3 \leq h_{norm} \leq 2/3 \\ group\ 3, & h_{norm} > 2/3 \end{cases}$$

**Cluster (K-means)**

- Set k = 2 or 3 (group)
- Two dimension : height and $\kappa_e$
- Order of group is based on order in the original snowpack

**Averaging**

**h-average**
Density, SSA and temp
Weight = $h$

**$\tau$-average**
SSA and temperature
Weight = $\kappa_e h$
Density
Weight = $h$

**Radar equivalent snowpack: k = 2 or 3**
k layers x 4 properties

18. Line 184: re-defines RMSE, why? It's not used in that section again.
No but it is used later in the paper multiple times.

19. Line 195: The last sentence in the paragraph (starting with "The difference in backscatter was estimated…") is unclear.
The sentence was modified as follows:
*The difference in backscatter was estimated between the reference simulation and the transparent simulation to estimate how much the internal interfaces are contributing to the total backscatter.*

20. Line 206: "… layers classified as 2 at …". Please make it more clear, e.g.: "classified/assigned as/in/under group/category 2.
We added group #2
*The two layers classified as group #2 …*

21. Line 210: "… with averaging using the h_i method" This is referred later as h_i-average method? Please use consistent terms for clarity.
We modified the term to be always h-average and \tau-average

22. Line 211: "(1 to 3)" -> "(from 1 to 3)"
We made the modification

23. Line 213: Please clarify sentence "Using a cluster is not always better …". E.g. "… Using a K-means clustering method is not …"
The sentence was modified as follow:
*Using the K-means clustering method is not always better than equal grouping*

24. Line 214: "\tau-method". This is later referred to as \tau-average? Please use consistent terms for clarity.

Yes, see comment 21

25. Line 216-217: "Figure 4 shows biases … as a function of snow properties". Seems a bit awkward sentence, add "methods" after mentioning the two methods?

We modified the sentence as follow:

*Figure 4 shows biases for the 3-equal and 3-cluster with h-average method as a function of snow properties*

26. Figure 3. is a bit cluttered with the number of time series. Is it necessary to show the differences in its own plots? How would the results look in a scatterplot?

We modified Figure 3 by only showing the different grouping and averaging methods for 3 layers.

[Figure]

*Figure 3. Backscatter time series for reference simulations and different 3-layer grouping and averaging methods. The simulations for the 2013-2014 season at WFJ, SAP and TVC are shown*

27. Line 231-233 "The 3-equal with \tau-average achieved similar performance to the …" This seems like an important result. Why is it not mentioned in the conclusions? Is it not so that due to its simplicity, the 3-equal grouping is more reliable compared to K means?

The 3 equal grouping is not more reliable than the K-means. Only that the averaging is a more important factor than the grouping in preserving the backscatter. We also added this in the conclusion.

*We found that the averaging method was more important than the grouping method in preserving the backscatter.*

28. Could Table 2 be replaced with a graph representing the information, and the Table moved to appendix? This could help to convey the message more clearly.

We decided that the Table was the best option to show the result for all sites, multiple seasons and methods.

29. Section 3.2. is interesting but on the other hand the manuscript is quite long. Is this section necessary for the main point of the manuscript? If it is, then some technical comments: "> 10 GHz, up to 30 GHz" please use words instead of symbols, e.g. "from 10 GHz, up to 30 GHz". Same comment for "> 40 GHz". Sentence "…3-equal with h-average because…" add "method" after h-average.

We think this section is important enough to be in the paper. We removed the symbol in this section.

30. Table 3: Change column name "layering" -> "number of layers" and remove repetition of "layer" from the data rows. Similarly, column name "\sigma_0 computation time" could include the unit (s) and avoid repeating it on the data rows. In my opinion the grouping times and associated \sigma_0 computation times (the last 4 numbers in the table) could be given in the caption of the table to reduce its size and make it clearer.

We added all your proposed change. This sentence was added in the caption. We also moved this section to the appendix.

*The backscatter computation time for 3 layers is 0.15 s and 0.12 s for 2 layers. The grouping method takes 0.005 s.*

31. Line 274: Similar to previous comment, sentences "… while preserving their microwave scattering behavior …" expression "preserving" is perhaps too strong. Please consider another expression, e.g. "aiming to preserve".

We feel like "preserving the scattering behavior" reflects that. We are not saying we are conserving scattering or preserving the scattering. However, we added "nearly impact the scattering behavior" in the previous paragraph.

32. The last paragraph in conclusions is a good addition. However, I found the paragraph too vague. The application of the method to SWE retrievals is one of the main motivations of the study and it is mentioned on many occasions in the manuscript. Therefore, I think it requires more attention and concrete detail. Also, I think this part should be moved to discussions and summarized in conclusions in shorter format.

We added this paragraph on SWE retrievals implication in a new section (3.3) of discussion.

*In SWE retrieval applications, the number of variables that need to be optimized plays a crucial role in determining the accuracy and efficiency of the retrieval process. One of the most significant advantages of adopting a radar equivalent snowpack representation is its ability to reduce the number of optimized variables without substantial loss of information. Now, an important choice still has to be done on whether 2 or 3 layers is best. We saw that backscatters from a 2-layer snowpack are slightly degraded compared to using a 3-layer snowpack (Table 2). Despite this small degradation with simplifying into 2 layers, retrieval applications could still benefit in terms of computational efficiency and reduced solution space, which can be advantageous for operational or large-scale applications. However, simplifying into 3 layers would offer a better representation of snowpacks across all climates.*

*In a Bayesian retrieval, calculating the mismatch term involves running a radiative transfer model and then optimizing the resulting posterior parameters. Using the radar equivalent snowpack would imply that the optimization is performed for a snowpack represented by a reduced number of layers, rather than its full complexity. Concurrently, the prior distributions for snow properties, which are sourced from SVS2, are also reduced to this number of layers. This consistent approach ensures that both the forward modeling for the mismatch term and the prior information is based on a comparable, simplified structural representation of the snowpack.*